# Integrating Fused Deposition Modeling and Melt Electrowriting for Engineering Branched Vasculature

**DOI:** 10.3390/biomedicines11123139

**Published:** 2023-11-24

**Authors:** Quinn S. Thorsnes, Paul R. Turner, Mohammed Azam Ali, Jaydee D. Cabral

**Affiliations:** 1Department of Oral Rehabilitation, School of Dentistry, University of Otago, Dunedin 9054, New Zealand; quinn.thorsnes@gmail.com (Q.S.T.); azam.ali@otago.ac.nz (M.A.A.); 2Department of Microbiology & Immunology, University of Otago, Dunedin 9054, New Zealand; paul.turner@otago.ac.nz

**Keywords:** polylactic acid, polycaprolactone, melt electrowriting, additive manufacturing, 3D printing, scaffold, tissue engineering blood vessels

## Abstract

We demonstrate for the first time the combination of two additive manufacturing technologies used in tandem, fused deposition modelling (FDM) and melt electrowriting (MEW), to increase the range of possible MEW structures, with a focus on creating branched, hollow scaffolds for vascularization. First, computer-aided design (CAD) was used to design branched mold halves which were then used to FDM print conductive polylactic acid (cPLA) molds. Next, MEW was performed over the top of these FDM cPLA molds using polycaprolactone (PCL), an FDA-approved biomaterial. After the removal of the newly constructed MEW scaffolds from the FDM molds, complementary MEW scaffold halves were heat-melded together by placing the flat surfaces of each half onto a temperature-controlled platform, then pressing the heated halves together, and finally allowing them to cool to create branched, hollow constructs. This hybrid technique permitted the direct fabrication of hollow MEW structures that would otherwise not be possible to achieve using MEW alone. The scaffolds then underwent in vitro physical and biological testing. Specifically, dynamic mechanical analysis showed the scaffolds had an anisotropic stiffness of 1 MPa or 5 MPa, depending on the direction of the applied stress. After a month of incubation, normal human dermal fibroblasts (NHDFs) were seen growing on the scaffolds, which demonstrated that no deleterious effects were exerted by the MEW scaffolds constructed using FDM cPLA molds. The significant potential of our hybrid additive manufacturing approach to fabricate complex MEW scaffolds could be applied to a variety of tissue engineering applications, particularly in the field of vascularization.

## 1. Introduction

Additive manufacturing (AM) includes a variety of manufacturing technologies and techniques that produce three-dimensional (3D) objects through the addition of material layer by layer. These AM technologies have been applied to tissue engineering such that custom, patient-specific, and anatomical geometries, each with their own advantages in specific scenarios, can be created. Of particular interest to this research are melt electrowriting (MEW) and fused deposition modelling (FDM) [1].

MEW has seen growing interest within the tissue engineering realm due to its ability to precisely fabricate microfibrous scaffolds with controllable, biomimetic, and microarchitectural detail. Similar in concept to electrospinning, but without the use of toxic organic solvents, MEW uses an electrical potential difference between the negatively charged biomaterial, polycaprolactone (PCL), and the positively charged printing platform to directly write stretched strands in the micrometer range [2,3].

MEW’s use of an external electric field also poses a challenge when compared to some other AM technologies, as the electric field continues to exert force on deposited sections. For example, overhanging structural features are challenging to fabricate without specific considerations [4]. It can also make previously placed parts of the scaffold act as antennae for subsequent layers and lead to an uneven distribution of material across the scaffold. This challenge is most relevant when designing scaffolds that have hollow regions or overhanging structures, such as scaffolds for growing replacement blood vessels. With these scaffolds, a hollow lumen must be produced reliably and without any lingering obstructions that could diminish the effectiveness of the grown vessel. These scaffolds should, ideally, also support complex, branching lumina of varied orientation and diameter [5,6].

One way to address this challenge is to include a mold to support the MEW scaffold during construction. Previous groups demonstrated that polyvinyl alcohol (PVA) can be useful as a mold material due to its water solubility [1]. This has the distinct advantage of being able to be removed without mechanically interacting with the scaffold, which could be desirable for intricate support structures or for delicate scaffolds. The disadvantage is that these molds are single-use.

It would be desirable to have a mold that is both reusable and able to positively interact with the electric fields inherent to MEW, while still being easily printed in custom forms. The use of FDM to create custom forms has seen a rapid growth in the consumer market in recent times. Here, a roll of filament is melted immediately before being extruded through a nozzle and deposited on the printing surface layer by layer. Common materials for FDM include polylactic acid (PLA) and acrylonitrile butadiene styrene (ABS), although the variety of custom materials is increasing as the technology gains more widespread acceptance in the consumer market. An additional material compatible with FDM of note for this research is PLA that has been made electrically conductive with the addition of graphene (cPLA) [7]. An electrically conductive material would interact with the electric field and attract the filament. Therefore, the aim of this research was to use cPLA to create conductive, branched mold vessel halves using FDM. MEW halves were then fabricated on top of the FDM molds, and the MEW scaffolds were carefully removed from the molds post-print and then melt-melded together to create hollow, branched constructs. The cPLA mold’s effects on the fabrication of MEW prototype scaffolds were determined along with the MEW scaffolds’ mechanical and biological properties for potential tissue engineering applications.

## 2. Materials and Methods

### 2.1. Materials

Polylactic acid (PLA) and acrylonitrile butadiene styrene (ABS) are the most common thermoplastics used with FDM printers. As the technology gains wider adoption, the range of compatible material grows. For this research, we used PLA with the addition of graphene to make it electrically conductive (cPLA) [7]. The cPLA used in this research had resistivity similar to that of seawater [8,9]. Polycaprolactone (PCL) was chosen for these experiments as it is a synthetic polymer that both is compatible with MEW and was FDA-approved as a material for Class 3 medical devices, including long-term implants [10,11]. PCL was also shown to be a good base for mixing with other materials [2,12,13]. Medical grade PCL (CAPA 6506; Perstorp Holding AB, Sweden) and cPLA (Jaycar Electronics; Wellington, New Zealand) were used. PLA (3D SUPPLY COMPANY LIMITED; Auckland, New Zealand).

### 2.2. Mold Design

A variety of molds were designed through the use of the CAD software, SOLIDWORKS (ver. 2019) [14]. The molds ranged in complexity from simple cubes and cylinders to complex branching structures. A variety of shapes were chosen to test the range of usable mold shapes and how the optimal printing parameters changed for different molds. The complex molds were designed to be branching structures with variable radii to resemble structures similar to arteries. Figure 1 shows the schematic diagram for these branching molds. The pictured mold radius varies from 5 mm at the base to 2 mm at the furthest branches, with a 2 mm padding beneath to allow for extra material for the later fusion of the scaffolds.

A set of simpler hemi-cylindrical molds were also prepared to obtain simpler scaffolds for analysis. A set of three half cylinders was designed with variable offsets from the base and separations. Figure 2 shows one such configuration with 2 mm wide humps that are 2 mm off the base and 5 mm apart. 

Similarly, two hollow cylinders were designed. The larger of the two had a 12 mm outer diameter and a 9 mm inner diameter and was 8 mm high. The smaller of the two had an 8 mm outer diameter and a 6 mm inner diameter and was 5 mm high. The motivation for the extra height of these molds was to further encourage adhesion to the mold, rather than to the glass surface below. 

### 2.3. Mold 3D Fabrication

The molds were fabricated using fused deposition modelling (FDM). Prior to fabrication, each mold design was tested for printability using standard PLA at a comparatively large layer height (0.3 mm). Natural PLA was used for this testing phase due to its reduced cost. Printability in this case meant no obvious deformations present in the printed model with the selected printing settings. The default, manufacturer-optimized settings for PLA (Table 1) on the Flashforge Creator Pro (Figure 3A) were used first. The settings were found in the slicing software from the same company, FlashPrint v3.10.0 [15]. Optimum print parameters were selected based on surface roughness, mechanical strength, and print fidelity. Print fidelity was checked via measurement using digital calipers.

The printer was selected for its dual extruders, which allow for multiple materials to be combined in a single print. Each extruder brass nozzle had the standard 0.4 mm internal diameter. Figure 3B shows a mold removed from the printer prior to completion, as well as the infill pattern used.

The models were then printed again, using cPLA at a lower layer height for increased detail (~0.12 mm) in any parts where this would affect the surface roughness. The initial print settings for cPLA were the same as for natural PLA. This worked well, and no further optimization was required. To improve the reliability of the prints, the cPLA molds were printed on a raft of natural PLA. As specified by the manufacturer, cPLA had a resistance of 300 kΩ per linear meter [8]. The filament was 1.75 mm in diameter, which corresponded to a resistivity of 0.7215 Ω m, as determined by Equation (1), where *R* is the resistance in Ohms, *ρ* is the resistivity in Ohm meters, *l* is the length in meters, and *A* is the cross-sectional area in square meters [16]. As a point of comparison, copper has a resistivity of 16.8 nΩ m, and sea water has a resistivity of 0.223 Ω m at 35 g/kg salinity [9,17].

Equation (1)—General resistivity formula for objects of uniform cross section:(1)$$R=\frac{\rho l}{A}↔\rho =\frac{RA}{l}$$

### 2.4. Melt Electrowriting 3D Printing

The molds were then taken to the GeSiM Bioscaffolder 3.1 (Radeberg, Germany) configured for melt electrowriting (MEW) and placed on the glass printing surface atop a high-voltage platform. The cartridge nozzle was moved downwards to the desired printing location until it was a few centimeters above the surface. This allowed the mold to be easily centered underneath the nozzle (Figure 4).

With the mold in place, a scaffold model was designed to be printed over the mold. In most cases, this was a rectangular cuboid. Hexagons and circles were also used when they would be more suitable for the specific mold. The preprinting procedure involved turning on the pressure and voltage and then touching the strand to the build plate and pulling the nozzle away. Then, the print was set to execute. This allowed the filament to reach a stable flow rate and avoid a large blob of material at the edge of the scaffold [20]. For comparison, the same print settings were also tested on both glass molds and a flat glass surface. The glass molds were made from test tubes and glass stirring rods bisected along their length and glued to a section of the glass printing surface (Figure 5). The plate with the attached molds was always placed in the same location on the high-voltage platform with the aid of reference marks, for a consistent printing location. 

The variables for melt electrowriting were the material temperature, the nozzle internal diameter, the accelerating voltage, the nozzle translation speed, the vertical nozzle jump per layer, the infill separation, the infill angle change, the extrusion pressure, and the distance of the nozzle to the printing substrate. The nozzle diameter was kept at a constant length of 45 µm, the material temperature was kept at 90 °C, and the nozzle jump per layer was 40 µm. All other parameters were varied, one at a time, in order to optimize the consistency of coverage of the strands on the mold and to minimize the pore size [2]. Table 2 shows the most desirable settings determined for MEW.

In order to construct hollow structures, two complimentary MEW halves were placed on a temperature-controlled platform set to 85 °C for one second and then pressed together and cooled at room temperature. Figure 6 shows the final intact 3D hollow structure.

### 2.5. Scaffold Analysis

#### 2.5.1. Pore Properties

The scaffolds were primarily analyzed using stereological techniques. These were used to estimate both the size of the pores and the fraction of the scaffold that the pores occupied. The volume fraction of the pores is also referred to as the porosity. The first step involved taking detailed pictures of a single scaffold layer using a microscope. Samples were prepared by printing out a single layer from the GeSiM robotics software (version 1.15.0.3689, Radeberg, Germany) onto each of the substrates under identical conditions. Porosity could be independently measured using relative density calculations. To obtain images for stereological analysis, the single-layer samples were placed between two microscope slides to flatten them. Microscope images were obtained and imported into FIJI (1.53 t) [21]. A grid was generated with random offset in the *x* and *y* dimensions and grid cell sizes of 10 mm^2^. The first stereological measure was performed by counting how many of the grid intersections intersected with free space relative to the number of grid intersections that fell within the image of the sample. This provided an estimate of the porosity of the sample, according to Equation (2) [22], where *V_V_(Y, ref)* is the volume fraction of *Y*, the element of interest, in the reference volume, *P(Y)_i_* is the sampled probability of hitting *Y* in sample *i*, and *P(ref)_I_* is the sampled probability of hitting *ref* in the same sample. The grids were regenerated, and sampling was repeated four times to improve confidence in the results.

Equation (2)—Stereological volume fraction estimator based on multiple samplings:(2)$$\hat{V_{V}}\left(Y,ref\right)=\frac{\sum_{i=1}^{m}{P(Y)}_{i}}{\sum_{i=1}^{m}{P(ref)}_{i}}$$

To obtain an estimate of the pore size, the width of each pore was measured each time the previously mentioned crosses intersected free space. Before this, an angle to the horizontal was randomly determined. All width measurements were made at precisely this angle and passed through the intersecting gridlines within the open space. The mean of these lengths was used to estimate the pore size, and the pore volume was estimated using Equation (3) [23].

Equation (3)—Stereological mean pore volume estimator:(3)$$\hat{\overset{-}{v_{V}}}=\frac{\pi }{3n}\sum_{i=1}^{n}l_{\left(0,i\right)}^{3}$$
where $\hat{\overset{-}{v_{V}}}$ is the estimate of the average pore volume, *n* is the number of pores measured, and *l_0,i_* is the measured pore size for the sampled pore.

The porosity of the samples was also measured using relative density to independently verify the results of the stereological estimation. Samples of 5 mm × 8 mm × 8 mm were made by using identical melt electrowriting parameters. The final samples were then trimmed to a cuboid shape with a razor and a cutting mat, and their dimensions were then measured with Vernier calipers. The samples were then weighed on a microgram-sensitive scale. The measurements were used to calculate the density of the samples, which was then compared to the density of solid PCL. The relative density of the scaffold with respect to that of PCL provided the volume fraction of PCL and, thus, the volume fraction of the pores as 1 minus the volume fraction of PCL.

#### 2.5.2. Mechanical Strength

The mechanical strengths of the scaffolds were measured directly using dynamic mechanical analysis (DMA). DMA tests provide information about both the viscous (liquid-like) and the elastic (solid-like) properties of a material. Given that biological conditions involve varying blood pressure and other stresses, the dynamic properties are useful for validating the scaffolds against the material strength of human arteries. Samples were prepared by cutting large, thick scaffolds with a fresh razor into four cuboid samples with dimensions of approximately 8 mm × 8 mm × 5 mm. These were the same samples that were used for the relative density calculations. The actual dimensions of each sample were carefully measured with digital Vernier calipers before the samples were placed in a Netzsch DMA 242 E Artemis (NETZSCH-Gerätebau GmbH, Selb, Germany) machine calibrated and configured for compression testing. The samples were then subjected to cyclic compressive stress at frequencies ranging from 0.1 to 3.33 Hz at 37.5 °C, with each batch of tests repeated for each sample several times over the course of 50 min. The atmosphere for these tests was gaseous nitrogen with a flow rate of 50 mL per minute.

This process was repeated for each sample such that they were tested with compression both parallel to and normal to the orientation of the layers. This repetition in different orientations was to test if the mechanical strength depended on the orientation of the layers within the scaffold. The data were then converted into CSV (comma-separated value) format using the Proteus Analysis version 6.1 software.

#### 2.5.3. In Vitro Cell Studies

The scaffolds for these experiments were branching scaffolds printed at the scale shown in Figure 1. All scaffolds were sterilized by washing with a solution of 70% ethanol for 10 min under shaking and then UV-irradiated for 30 min. The scaffolds were then rinsed with sterile phosphate-buffered saline (PBS), and some scaffolds were soaked in 0.1% gelatin in PBS for 24 h. The incorporation of gelatin can enhance cell–scaffold interactions by mimicking cell-adhesive ECM components that affect cellular behavior such as cell proliferation and migration [24]. The geometry and size of the scaffolds used are depicted in Figure 1. All scaffolds were then rinsed in PBS under shaking for 10 min in sterile conditions. 

Biocompatibility was investigated in vitro using cultured normal human dermal fibroblasts (NHDFs) obtained from the American Type Culture Collection (ATCC^®^, Manassas, VA, USA). The cells were grown in Dulbecco’s modified Eagle medium (DMEM) (Gibco™, Grand Island, NY, USA), containing 10% fetal calf serum and penicillin/streptomycin. The cells were routinely passaged at 80% confluence using TrypLE™ Express (Gibco™, København, Denmark) and seeded into new plates at a density of 5000 cells/cm^2^. Fibroblasts were chosen for their similar size to that of human endothelial cells (~20 µm). Prior to seeding, the cells were checked for viability under a microscope and counted using a hemocytometer. A total of 1.2 million cells were seeded directly on top of a scaffold, then completely submerged in the culture medium (using, approximately, 15 mL). The dishes were then incubated at 37 °C in an atmosphere containing 5% CO_2_ [25]. Every 3–4 days during the incubation, the scaffolds were inspected for growth, and approximately half of the culture medium was removed and replaced with fresh DMEM.

After one month, the scaffolds were removed and fixed with 5 mL of 4% paraformaldehyde at room temperature for 15 min, followed by washing in PBS. The scaffolds were then stained with phalloidin–Alexa Fluor™ 594 (diluted 1:1000, Invitrogen™, Carlsbad, CA, USA) to visualize the actin fibers and Hoechst 33342 (100 ng/mL, Themo Scientific™, Waltham, MA, USA) to stain the nuclei, in PBS overnight in the dark. Fluorescence microscopy images were obtained on an Olympus inverted IX71 microscope. Brightfield images were also taken to visualize the scaffolds. Composite images were created using ImageJ software, version 1.53u.

## 3. Results

### 3.1. Mold Fabrication

The default FDM parameters for PLA worked well for conductive PLA. The primary complication, affecting every print using cPLA, was that cPLA is extremely brittle. This necessitated feeding the filament into the printer by hand for the duration of each print. This made the fabrication of larger molds prohibitive. The difference in material stiffness of conductive and natural PLA allowed the conductive molds to be removed from the rafts by bending. We found that the 1 mm thick mold and the smaller cylinder underperformed (Figure 7). The 1 mm thick bump mold was too weak and brittle, leading to the bumps breaking off when the mold was moved. The smaller of the two cylinders (8 mm outer diameter) was almost unusable due to its low mass. This caused the mold to be knocked over and dragged along by the extruded filament in some of the test prints. Additionally, this mold was printed functionally as two concentric hollow cylinders with a wall thickness of 0.4 mm. This was likely an artefact of the specific design of the mold and the version of the slicer. 

### 3.2. MEW 3D Printing

Comparing the mold materials showed that the material tended to accumulate within the scaffold. For the glass molds, even coverage could be acquired reliably in low-voltage situations. In high-voltage tests, the material tended to accumulate towards the bottom of the mold and on the flat substrate below, as shown in Figure 8.

Similarly, a relatively even coverage could be achieved with the cPLA molds when the potential difference between the nozzle and the printing surface was lower (Figure 9A). As the electrical potential difference increased, it was observed that the material tended to accumulate at the top of the mold. For the hollow cylindrical molds, we took advantage of this tendency of the printed material to adhere to the top of the mold. This allowed for a completely hollow, porous, and tubular structure to be made directly (Figure 9B). It was observed that the scaffold tended to collect on the inner part of the mold and decrease in radius in subsequent layers. This was offset by having the designed model be slightly larger in both inner and outer radius than the mold itself.

The scaffolds could be easily removed from both the glass molds and the cPLA molds without force or damage to the scaffolds themselves. The tests with natural PLA molds, however, found that the PCL scaffolds were strongly adherent to the PLA molds, to the point that the scaffolds were broken upon removal. 

### 3.3. Scaffold Analysis

#### 3.3.1. Pore Properties

When looking at the scaffolds under a microscope, it was observed that the microstructure of the scaffolds varied with the height of the mold and with the material used for the mold. The scaffolds printed on sections of a glass mold further from the nozzle, for example, showed more whipping or secondary electrical instabilities [26] than those atop the mold, where a controlled square grid could still be achieved. The scaffolds printed onto cPLA showed tighter whipping than those on glass under the same settings. The strands were also observed to be marginally thinner, of approximately 30 µm compared to 35 µm. As can be seen in Figure 10, there was some variation in the thickness of these strands due to various factors, such as the strands doubling back on themselves.

Figure 10 shows views of single-layer samples. Multiple strands appear overlapping in both cases. These overlaps were caused by the strands being allowed to whip in their path during the print. This was an intentional design choice to produce a more organic geometry and allow the strands to be closer to each other. Figure 10B shows that the layers within the scaffolds on the cPLA mold seemed to have fused together more strongly.

The pore size and volume measurement results are shown in Table 3. Here, a clear difference in pore size can be seen when printing onto a conductive mold compared to flat glass. The print on glass was sampled twice to measure the variability in the results due to different sampling angles. The corresponding stereological estimates of porosity are shown in Table 4. It should be noted that the samples used for these tests were produced with the same parameters, except for the printing surface.

Thicker samples were then prepared for both the flat glass and the cPLA printing surfaces. In both cases, these were designed to be printed as 20 mm × 20 mm × 5 mm cuboids that could be cut to size for mechanical tests. Printing on glass resulted, instead, in what appeared to be flat-topped square pyramids (Figure 11A) with a base wider than that designed and a top that was narrower. The cuboids printed onto cPLA were not satisfactory, even when using a single mold at a time (Figure 11B). Only glass allowed obtaining usable samples. The resulting measurements of the relative density calculations can be seen in Table 5.

The more accurate relative density results correlated better with the original designs. The difference between the stereological estimates suggested oversampling of the scaffold during the stereological analysis. Predicted oversampling can be calculated by determining the actual chance of hitting a pore with the estimation by asserting that the estimation is based on the areas that include pores in all layers. The probability of hitting a pore on every layer, given the probability of striking a pore in any layer, is expressed by Equation (4) [27].

Equation (4)—Probability of all pores being open at a sampling location:(4)$$P\left(A\;and\;B\right)=P\left(A\right)P\left(B\right)\Rightarrow P\left(pore,n\right)=P{\left(pore,1\right)}^{n}\to n=\frac{log\left(P\left(pore,n\right)\right)}{log\left(P\left(pore,1\right)\right)}$$
where *P(pore, n)* is the probability of a random location hitting a pore in each of the *n* layers, and *P(pore, 1)* is the probability of hitting a pore in one layer. Solving this equation, if the actual porosity is 77.08% and the estimated porosity is 47.56%, we obtain an average of 2.87 layers being sampled in the glass model. Assuming a similar oversampling rate, this would suggest that the samples made on cPLA molds would be predicted to have an actual porosity of roughly 63.29%.

Figure 10 clearly shows that there were several overlapping strands in both cases. A qualitative analysis of the two images suggested that cPLA led to a similar or greater degree of oversampling in the samples produced for stereology.

#### 3.3.2. Mechanical Strength

The DMA test results, shown in Figure 12, indicated a directional dependence on the material stiffness. The vertical compressions tests, where the layers were pressed further into each other, showed a storage modulus around 1 MPa, while the horizontal compressions tests, where the stress lay within the plane of the layers, indicated a storage modulus of around 4.5 MPa. It was also seen in both tests that the loss modulus—the viscous part of the complex modulus, describing energy loss as heat—remained an order of magnitude smaller than the storage modulus in both cases, indicating that the scaffolds were mostly elastic.

In both cases, the trend indicated a rapid increase and then a slow decay, peaking around the same frequency. The plot also shows that the scaffolds were more resistant to stresses that lay within the plane of the layers by a factor of roughly 4, compared to stresses that were normal to this plane. These values are greater than those of human arteries (up to 2.7 kPa) but lower than those of scaffolds made with current synthetic graft materials such as ePTFE and PET [5,28].

#### 3.3.3. In Vitro Cell Studies

The cultured cells showed slow but consistent growth over the course of the four weeks of incubation. Brightfield microscopy showed that the NHDFs were most visibly attached to the strands on the periphery of the scaffold at an early stage. Observing cells within the volume of the scaffold was more challenging during this time, due primarily due to the thickness of the scaffolds. Living cells remained visible in the PCL scaffold pores after four weeks of incubation. Similar to what we observed in previously published work by our group for NHDFs growing on MEW PCL scaffolds in long-term culture, the cells were able to proliferate [13].

Figure 13 shows that NHDFs infiltrated the volume of the scaffold and preferred areas where the strands were particularly close together, thus reinforcing the idea that smaller pores are preferable for cell growth.

## 4. Discussion

The presence of graphene in cPLA significantly increased brittleness. While this did not impact the quality of the molds, the need for constant supervision made the production of large molds impractical. This problem could potentially be mitigated by using other, less brittle, conductive filaments in the FDM printer. Changing the base material of the mold could also improve scaffold adherence. The reason for the PCL scaffolds adhering to pure-PCL molds but not to cPLA molds is unclear, but poor cohesion of PLA–PCL–graphite compounds was observed previously [29]. The PCL used in these experiments did not stick to steel and aluminum instruments in the lab, thus metal may also be a desirable mold material, as could glass. Neither are obviously suitable for the rapid prototyping of custom molds for patients, as both glass and metal tend to require more expensive and robust equipment. However, the workflow for generating molds was found to be straightforward and reliable, once calibrated. The parametric definition of the molds, despite the apparent complexity of the branching examples, enabled modification with relative ease. The workflow itself is flexible and compatible with prior scans of the lumina in local regions. The only limitations to the design are the methods of fabrication. In this research, the nozzles on the FDM printers had an internal diameter of 0.4 mm, meaning no part of the molds could be smaller than this. Similarly, the slicer—software used to convert the 3D model file into instructions for the FDM—used in these experiments produced a gap in the wall in one of the prints. Changes to the settings or versions or using different slicers may solve this problem. 

The different behavior at higher voltage may be explained by the properties of the external electric field affecting the charged filament. In the case of glass molds, glass acts as an insulator; so, the place of lowest voltage difference was the glass plate, rather than the top of the mold. This was in contrast with what observed for the cPLA molds, which are conductive and thus have equal electrical potential throughout their volume. This means that the path of least resistance would, in most cases, be at the top of the cPLA molds due to the nozzle needing to be three millimeters above the mold to have room to stretch to the desired diameter [2]. For this reason, experimentation with nozzles of finer diameter could yield interesting results. Refinements to the input models for the MEW printer could also further optimize this process. The reason why on glass we obtained truncated square pyramids rather than cuboids when control over the strand was relaxed, may be understood by considering probability distributions. For each location a strand is directed to, it will have a random deviation from that point. Points randomly distributed about a central point form a Gaussian distribution. When several Gaussian distributions are superimposed, they form a taller, flatter curve, as shown in Figure 14, that resembles the observed truncated square pyramid.

While the estimates for stereology were poorly correlated with the results from the relative density analysis, they were still useful for estimating the properties of the pores themselves. Scaffolds with simpler shapes were easier to work with, whereas the full-sized branching scaffolds were too thick to allow distinguishing individual layers, and the curved shape limited the depth of focus. This is obvious in Figure 13 and even in the single-layer prints on glass where the prints (Figure 10) show overlapping strands, which is where the main oversampling is expected to originate. As previously mentioned, the relative density calculations were more accurate for overall porosity. However, they could not be used to determine the size and shape of the pores. Given the oversampling of the scaffold in the first instance, it seems likely that the pores were larger than the initial estimate. It may be possible to predict a correction factor based on the comparison between the average pore volume and the pore size based on the calculated oversampling rate of 2.87 [5].

Like the porosity estimates, simple shapes were required to test the bulk properties of the material. While directly testing the scaffolds with more complex shapes would have been ideal, it was impractical. The tests on simple shapes can instead be used to calibrate simulations of more complex geometries with confidence. In this manner, future designers using material with similar mechanical behavior to PCL can test more elaborate biological shapes with reasonable confidence before printing [30]. The difference between the vertical and the horizontal compressive stiffness is relevant to the forces in blood vessels that are both radial and circumferential as the vessels try to expand. The higher strength in the plane could result in a reduced risk of failure and prevent one area being particularly weak. Additionally, the scaffolds were stiffest when stresses were applied at frequencies of 1–2 Hz or 60–120 beats per minute, the same ranges as those of the typical human heartbeat. 

Although PCL is not a strongly adherent substrate for cells, the presence of NHDFs on the scaffold after one month indicated that there were no residues introduced in the manufacturing process that impeded growth [13]. Cell adherence could be improved with more bioactive materials [11,13] or additives in future investigations [2]. The high mechanical strength measured thus far indicates there should be little concern for the weakening of the scaffolds through the addition of natural polymers with higher biocompatibility [2]. Similarly, previous work using chitosan additives in PCL demonstrated the importance of the pore size [11,25]. In order to improve the biological properties of MEW scaffolds produced from PCL, combinations with other polymers to create composite structures were shown to improve the development of vascularized tissue-engineered constructs. For example, Shahverdi et al. used PCL composite scaffolds to successfully grow endothelial cells on their surface [31]. Additionally, other manufacturing techniques, such as electrospinning [32], that are appropriate for large-scale production should be compatible with these manufacturing techniques and materials. Confirming this in future work would be an important step in establishing the viability of cPLA molds to expand the repertoire of scaffold shapes produced using the MEW technology.

Engineering 3D tissues is a favorable methodology to restore or replace damaged or diseased tissues, but a major drawback is the lack of vascularization strategies available to sustain thick constructs [33]. Existing scaffold fabrication methodologies, such as 3D printing, solution electrospinning, and MEW, among other technologies, have been implemented to address this need over the past twenty years [34,35,36,37,38,39,40,41,42,43,44]. In order to generate scalable, more complex 3D structures, researchers have employed hybrid fabrication approaches that combine at least two additive manufacturing technologies to launch new prospects in manufacturing [45,46]. Each of these fabrication technologies have specific advantages and limitations and can be used together in complementary ways to create novel biomimetic structures. Previous work used rotating mandrels to create tubular electrospun scaffolds to mimic the size and scale of vessels [47,48,49]. Scaffold-based techniques are particularly advantageous, as they generate highly porous morphologies which are customizable with the modern manufacturing techniques [50]. Others investigated the use of shape memory materials [51] or integrated 4D printing to increase the shape variety using FDM [52]. Other conducting polymers as FDM molds for MEW, such as poly(3,4-ethylene-dioxythiophene), polypyrrole, and polyaniline [53], could be explored to improve the hybrid technology even further and potentially broaden its biomedical applications. Using merged techniques such as those demonstrated herein, featuring FDM-generated 3D cPLA molds as substrates for MEW to create porous and branched vascular constructs, is a promising strategy combining beneficial material properties and AM technologies in the design methodology.

## 5. Conclusions

This research showed the potential of combining multiple additive manufacturing technologies to diversify and increase the complexity of bioscaffold shapes and microarchitectures. Herein, we reported for the first time the tandem use of fused deposition modelling to create reusable cPLA molds and MEW directly on top of these molds, to construct complex microfibrous scaffolds featuring complex branched vascular designs with hollow lumens. A mechanical analysis showed the scaffolds exhibited anisotropic strength ranging from 1 to 5 MPa, which is greater than that of 2.7 kPa of some arteries [5]. A visual analysis found that focusing the electric field with a conductive mold decreased both the size of the pores and the thickness of the strands compared to printing on glass, resulting in a slightly reduced porosity. Using conductive polylactic acid to produce MEW molds was particularly advantageous, as direct writing control was maintained, along with ease of scaffold removal. A straightforward heat melding of duplicate MEW halves allowed for the easy construction of a 3D tissue-engineered vascular construct. Our hybrid methodology was able to successfully fabricate more complex biomimetic geometries when compared to MEW alone, which, in addition, displayed appropriate biological compatibility, as evidenced by the growth of NHDFs. Future work should involve the exploration of the use of other conducting polymers as FDM molds for MEW scaffold production to produce human-scale tissue constructs featuring appropriate structural integrity. The combined integration and/or functionalization of bioactives with PCL is key to ensuring high cell viability and differentiation to obtain appropriate tissue types.

## Figures and Tables

**Figure 1 biomedicines-11-03139-f001:**
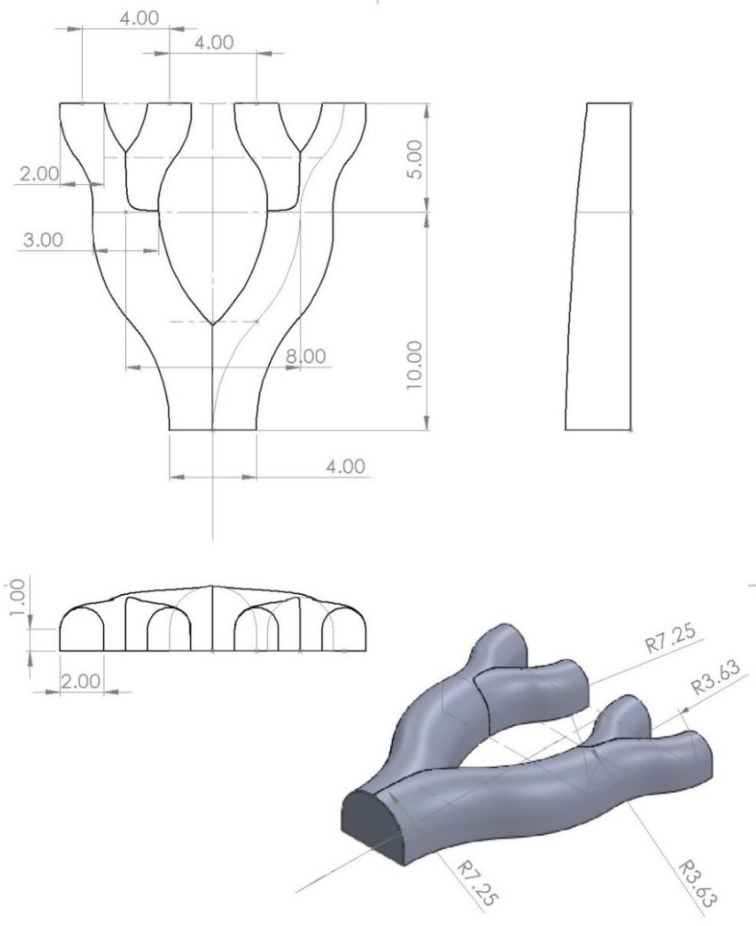
Schematic diagram of a branching mold. All lengths are in millimeters.

**Figure 2 biomedicines-11-03139-f002:**
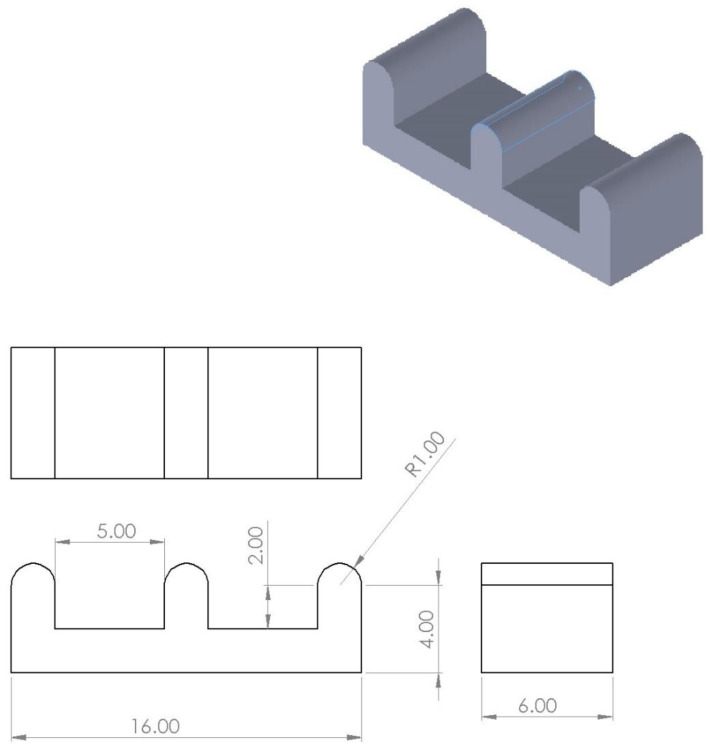
Diagram of a half-cylinder mold. All measurements are in millimeters.

**Figure 3 biomedicines-11-03139-f003:**
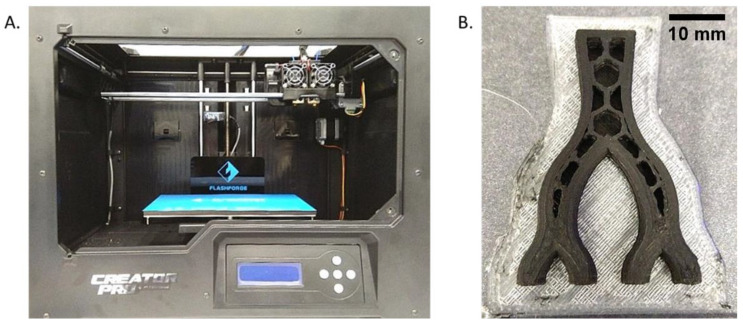
Flashforge Creator Pro dual-extrusion FDM printer (**A**), alongside a partial print of a cPLA mold (**B**).

**Figure 4 biomedicines-11-03139-f004:**
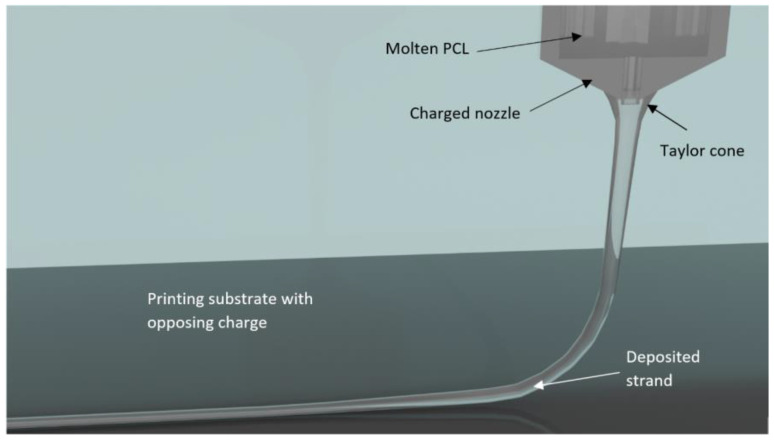
Schematic of the key features of melt electrowriting on a glass substrate made in Blender 3.4 [18,19].

**Figure 5 biomedicines-11-03139-f005:**
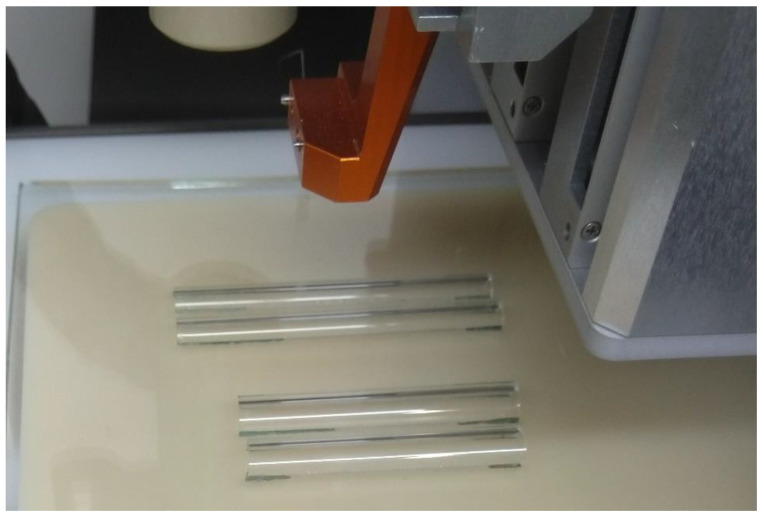
Two pairs of glass half-cylinder molds. One pair was 7 mm in diameter, and the other was 9 mm in diameter; they were placed on top of the MEW conducting platform.

**Figure 6 biomedicines-11-03139-f006:**
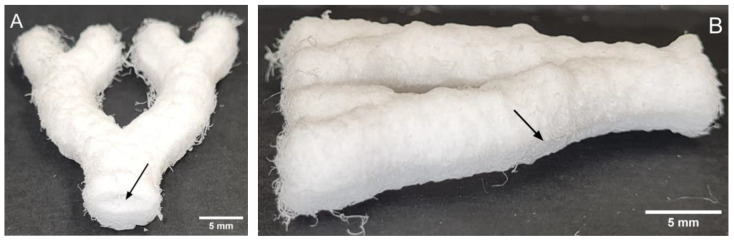
Final intact 3D hollow structure. (**A**) Arrow indicates pore opening. (**B**) Arrow indicates heat-melded longitudinal seam.

**Figure 7 biomedicines-11-03139-f007:**
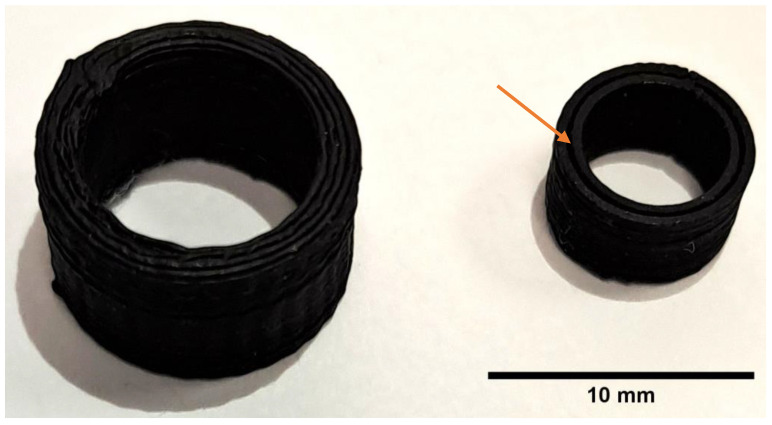
Two hollow cylinder molds. The arrow indicates a gap in the ring in the smaller mold.

**Figure 8 biomedicines-11-03139-f008:**
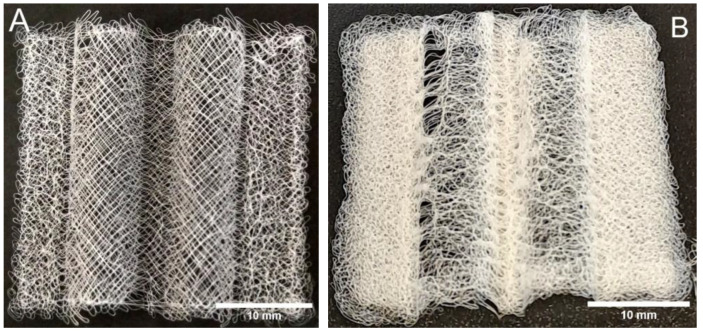
Higher fidelityprinting onto glass molds at low voltage (**A**) compared to high voltage (**B**).

**Figure 9 biomedicines-11-03139-f009:**
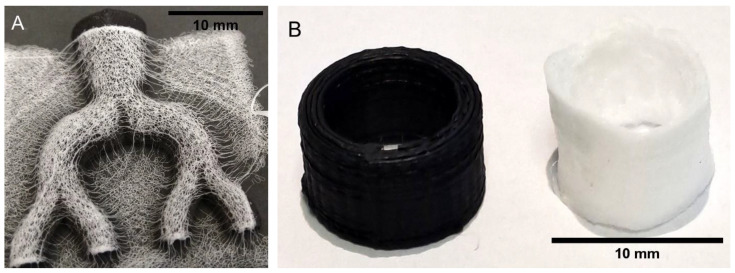
Partially controlled coverage on a cPLA mold (**A**) and selective accumulation on top sections (**B**).

**Figure 10 biomedicines-11-03139-f010:**
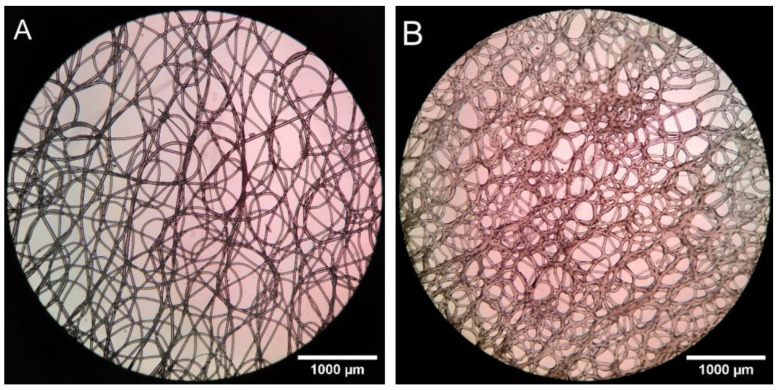
View of single-layer scaffolds printed on glass (**A**) and on a cPLA mold (**B**).

**Figure 11 biomedicines-11-03139-f011:**
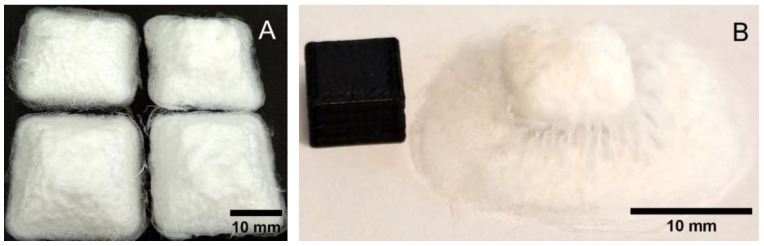
Printing thick square models onto flat glass (**A**) and cPLA (**B**).

**Figure 12 biomedicines-11-03139-f012:**
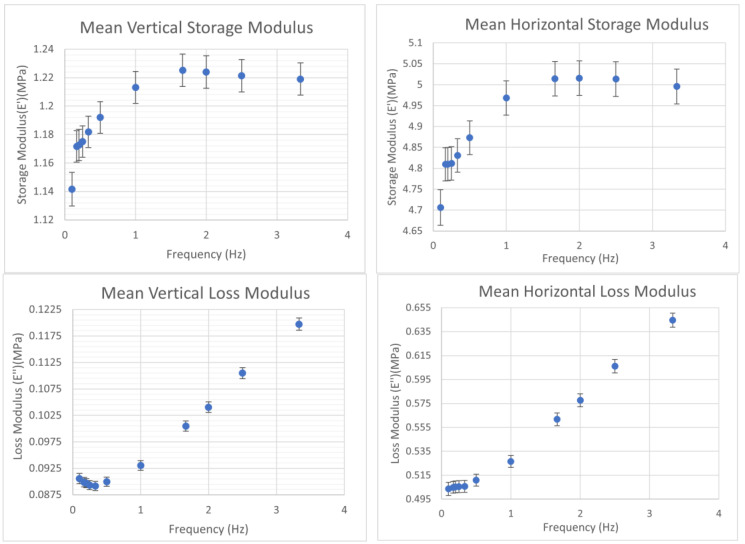
Plots of the time-averaged elastic moduli for the two orientations (vertical and horizontal) of the printed scaffolds with associated standard errors.

**Figure 13 biomedicines-11-03139-f013:**
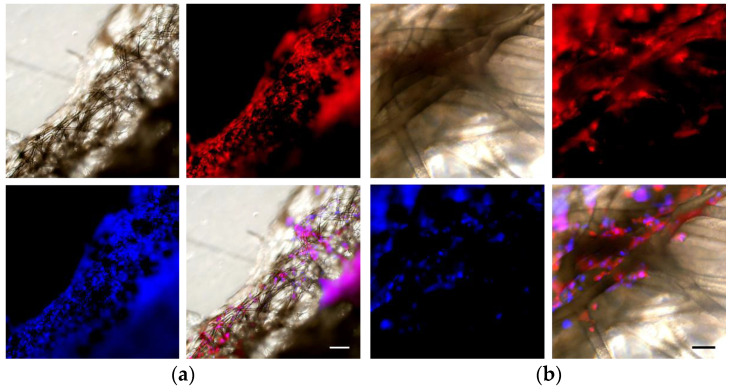
Fluorescent microscopy images of a scaffold with NHDFs after 4 weeks of growth. Clockwise from bottom left: NHDF nuclei stained with Hoechst 33342; brightfield image of the scaffold; NHDF actin fibers stained with phalloidin–Alexa fluor-594; composite of the previous three images overlaid. Scale bar is equal to 200 µm in (**a**) and to 50 µm in (**b**).

**Figure 14 biomedicines-11-03139-f014:**
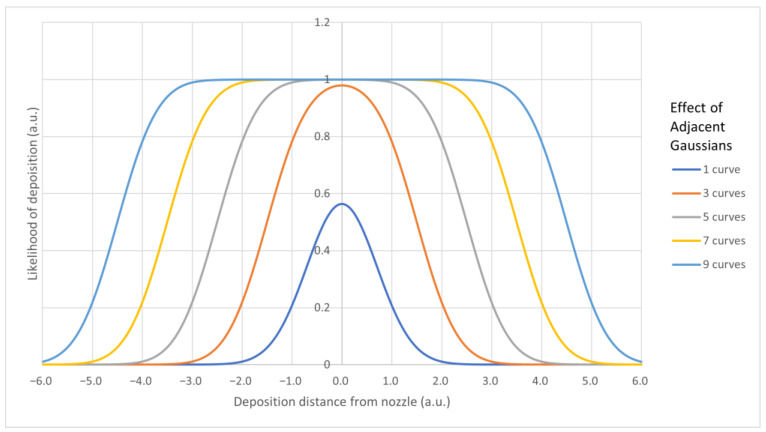
Plot of several Gaussian curves with unit spacing.

**Table 1 biomedicines-11-03139-t001:** Optimized FDM printing parameters.

Print Speed (mm s^−1^)	Nozzle Temperature (°C)	Bed Temperature (°C)	Infill (%)
70	205	50	30, hexagonal

**Table 2 biomedicines-11-03139-t002:** Optimized MEW printing parameters.

Strand Separation (mm)	T_Cartridge_ (°C)	Voltage (kV)	z-Spacing (mm)	Nozzle Speed (mm s^−1^)	P_Cartridge_ (kPa)	Infill Angle Change (°)
0.4	90	20	15	20	20	85

**Table 3 biomedicines-11-03139-t003:** Mean pore size and volume estimates from stereology.

Pore Statistic	Median Size (µm)	σ_size_ (µm)	Median Volume (nL)	σ_volume_ (nL)
Glass sampling 1	170	113	5.14	24.25
Glass sampling 2	165	110	4.72	24.06
Conductive Branch	115	71	1.39	10.28

**Table 4 biomedicines-11-03139-t004:** Estimated porosities from the stereology measurements for the samples in Table 1.

Printing Surface	Median, Glass	σ_Glass_	Median, cPLA	σ_cPLA_
Pore Volume Fraction (%)	48.07	2.33	29.47	4.46

**Table 5 biomedicines-11-03139-t005:** Relative density calculation to determine the porosity of the PCL scaffolds printed on flat glass.

Mass (mg)	Volume (mm^3^)	Density (kg m^−3^)	Porosity (%)
82.0 ± 0.1	328 ± 1	250 ± 2	77.6 ± 0.5
62.4 ± 0.1	246 ± 1	254 ± 2	77.2 ± 0.5
58.8 ± 0.1	226 ± 1	260 ± 2	76.7 ± 0.5
79.5 ± 0.1	314 ± 2	253 ± 2	77.3 ± 0.5

## Data Availability

Data are contained within the article.

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
