# Peer review of "Integrating Fused Deposition Modeling and Melt Electrowriting for Engineering Branched Vasculature"

_biomedicines, 2023, doi:10.3390/biomedicines11123139_

Round 1

Reviewer 1 Report

Comments and Suggestions for Authors

Major Comments:

1. The authors mention that they want to use PLA instead of PVA because it is reusable. PVA is the standard because it can be easily removed with water, which would mean that the removal of the part from the mold would be crucial, yet the authors make no note of how intensive that was other than “minimal mechanical force”and if that was why half cylinders were used for branched structures instead of closed structures.

2. Additionally, it seems that an appropriate control for these experiments would have also been native PLA without the graphene so see how the graphene affects the MEW process. There is a brief mention of removing scaffolds from natural PLA but no images of the scaffolds in comparison or any real data or information on the difference between the two. More information should be added.

3. It is unclear what is meant by “In tests where the regularity in the scaffold was less important,” on page 10 line 319. In what tests was it important and which it was not as important and why the differences? 

4. For the pore size analysis the paper states “were produced with the same parameters except for 336

the printing surface, and these parameters were different to the final optimized parameters”. Why were the final optimized parameters not used for this test? An explanation should be added or the tests should be done with the optimized parameters.

5. The authors mention that this is to increase complexity but even the branched structure is shown as a flat mat on top of the underlying mold, there is no visual of how that would look in a “complex” 3d branched structure (which perhaps the title should be revised then as well). 

6.  In addition to the comment above, the pores seem too large for creating any type of vasculature as the cells won’t form a full monolayer and then the resulting structure would be leaky, so this seems like maybe not the best application for this technique in this paper in its current form. 

Minor edits: 

Page 1 line 45: there is a “.when necessary.” that doesn’t appear to go with either the sentence before or after it. 

On multiple pages, there are some “Error! Reference source not found. “ specifically in lines 88 and 94. Not sure if this was an issue with the way the file loaded

Figure 6 mentions an arrow but there is no visible arrow in the figure

Author Response

  1. The authors mention that they want to use PLA instead of PVA because it is reusable. PVA is the standard because it can be easily removed with water, which would mean that the removal of the part from the mold would be crucial, yet the authors make no note of how intensive that was other than “minimal mechanical force”and if that was why half cylinders were used for branched structures instead of closed structures. Response: The text has been modified to in Section 3.2, “…the scaffolds could be easily removed from the molds without force or damage to the scaffolds themselves.”
  2. Additionally, it seems that an appropriate control for these experiments would have also been native PLA without the graphene so see how the graphene affects the MEW process. There is a brief mention of removing scaffolds from natural PLA but no images of the scaffolds in comparison or any real data or information on the difference between the two. More information should be added. Response: The text has been modified to in Section 3.2, “…PCL scaffolds were strongly adherent to the PLA molds, to the point that the scaffolds were broken upon removal.” Due to the highly impracticable nature of this approach were the scaffolds retained known of their original structural design, natural PLA mold use was not pursued.
  3. It is unclear what is meant by “In tests where the regularity in the scaffold was less important,” on page 10 line 319. In what tests was it important and which it was not as important and why the differences? Response: This sentence was deleted and restructured stating the following, “Scaffold printed on sections of glass mold further from the nozzle, for example, showed more whipping or secondary electrical instabilities [new Ref [26] Kade, J. C., Dalton, P. D., Polymers for Melt Electrowriting.  Healthcare Mater.2020, 10, 2001232. https://doi.org/10.1002/adhm.202001232] than those atop the mold, where a controlled square grid could still be achieved.
  4. For the pore size analysis the paper states “were produced with the same parameters except for 336 the printing surface, and these parameters were different to the final optimized parameters”. Why were the final optimized parameters not used for this test? An explanation should be added or the tests should be done with the optimized parameters. Response: The final optimized parameters were used, and this sentence was deleted.
  5. The authors mention that this is to increase complexity but even the branched structure is shown as a flat mat on top of the underlying mold, there is no visual of how that would look in a “complex” 3d branched structure (which perhaps the title should be revised then as well). Response: An image of the complete 3D branched structure has been included [Figure 6].
  6. In addition to the comment above, the pores seem too large for creating any type of vasculature as the cells won’t form a full monolayer and then the resulting structure would be leaky, so this seems like maybe not the best application for this technique in this paper in its current form. 

 Response: The following was added to the Discussion section, “In order to improve the biological properties of MEW scaffolds produced from PCL, combining with other polymers to create composite structures have been show to improve development of vascularized tissue engineered constructs. For example, Shahverdi et al. used PCL composite scaffolds to successfully grow endothelial cells on their surface. (new Ref [31] Shahverdi, M., Seifi, S., Akbari, A. et al. Melt electrowriting of PLA, PCL, and composite PLA/PCL scaffolds for tissue engineering application. Sci Rep 12, 19935 (2022). https://doi.org/10.1038/s41598-022-24275-6]

Minor edits: 

Page 1 line 45: there is a “.when necessary.” that doesn’t appear to go with either the sentence before or after it. Response: This had been deleted.

On multiple pages, there are some “Error! Reference source not found. “ specifically in lines 88 and 94. Not sure if this was an issue with the way the file loaded. Response: These have been corrected.

Figure 6 mentions an arrow but there is no visible arrow in the figure. Response: Arrow has been added with figure now labelled as Figure 7 due to insertion of another image prior.

Reviewer 2 Report

Comments and Suggestions for Authors

1.       The message “Error! Reference source not found” appears in multiple places throughout the manuscript, so figure referencing is missing. Please, correct this.

2.       Following the previous comment, neither figure numbering nor table labeling follow a logical sequence. Please, revise.

3.       The writing of the manuscript should be thoroughly revised, as the current version of the document is riddled with syntax and grammar mistakes that impede reading comprehension.

4.       For every equation shown, please provide a description of each of the parameters involved.

5.       Line 401: The beginning of the sentence is missing. Please, revise.

6.       Section 2.5.3.: in the revised version of the manuscript, please address the following:

-          The title of this section should be replaced with “In vitro cell studies”, or “Cell adhesion studies”, or something along those lines. Evaluation of biocompatibility entails several types of comprehensive tests, which were not included in the study. Only cell adhesion was evaluated. Thus, the authors cannot claim that “biocompatibility” was assessed.

-          Accordingly, please revise the title for section 3.3.2.

-          Why was gelatin used to treat the samples prior to cell seeding? What was the rationale? Please, clarify this in the Materials and Methods section of the revised version of the manuscript.

-          Indicate the geometry and size of the samples that were used for the cell studies.

-          Provide a more detailed and clear explanation regarding how the cell seeding was conducted. From what is currently reported, it seems that the scaffolds were immersed in cell culture media containing cells. If that is the case, why was this approach selected, instead of direct seeding onto the scaffolds?

7.       Figure 12:

-          70% ethanol and UV irradiation are disinfection methods, but they do not ensure sterilization (killing 100% of microorganisms). Therefore, it would be extraordinarily difficult to maintain a “clean” culture of the scaffolds for 30 days, as reported by the authors. In this context, why was not cell-metabolic activity assessed?

-          Why was not higher magnification used for imaging? Given the above comment, and the fact that Hoechst can bind to damaged DNA as well as DNA of non-mammalian nature, taking images at higher magnification would have allowed to rule out contamination, and also it would have provided a clearer visualization of the integrity of cells attached to the scaffolds.   

8.       In section 3.3.2, the authors state that “This indicated that there was little to no cytotoxic effects introduced by the use of the cPLA molds and that any contaminants that were introduced could be mitigated with standard sterilization procedures.” This is highly inaccurate and misleading, given that cytotoxicity was not measured, and the scaffolds were not sterilized, only disinfected. Please, revise this statement in the updated version of the manuscript.

9.       Lines 518-519: Neither biocompatibility nor cell proliferation was measured/evaluated. Thus, these words should be removed from the text. Please, revise accordingly.  

Comments on the Quality of English Language

1.       The writing of the manuscript should be thoroughly revised, as the current version of the document is riddled with syntax and grammar mistakes that impede reading comprehension.

Author Response

  1. The message “Error! Reference source not found” appears in multiple places throughout the manuscript, so figure referencing is missing. Please, correct this. Response: Corrected.

  1. Following the previous comment, neither figure numbering nor table labeling follow a logical sequence. Please, revise. Response: Corrected.

  1. The writing of the manuscript should be thoroughly revised, as the current version of the document is riddled with syntax and grammar mistakes that impede reading comprehension. Response: Grammar and syntax mistakes have been corrected.
  2. For every equation shown, please provide a description of each of the parameters involved. Response: All equation parameter symbols are now explained.
  3. Line 401: The beginning of the sentence is missing. Please, revise. Response: Corrected with grammar and syntax of entire document.
  4. 6.       Section 2.5.3.: in the revised version of the manuscript, please address the following:

-          The title of this section should be replaced with “In vitro cell studies”, or “Cell adhesion studies”, or something along those lines. Evaluation of biocompatibility entails several types of comprehensive tests, which were not included in the study. Only cell adhesion was evaluated. Thus, the authors cannot claim that “biocompatibility” was assessed. Response: The title was replaced with “In vitro cell studies” as recommended.

-          Accordingly, please revise the title for section 3.3.2. Response: This was changed as well to match.

-          Why was gelatin used to treat the samples prior to cell seeding? What was the rationale? Please, clarify this in the Materials and Methods section of the revised version of the manuscript. Response: The following was added to the text, “Incorporation of gelatin can enhance cell-scaffold interactions by mediating cell-adhesive ECM components which effect cellular behavior such as cell proliferation and migration [new Ref:[24] Hwang PT, Murdock K, Alexander GC, Salaam AD, Ng JI, Lim DJ, Dean D, Jun HW. Poly(É›-caprolactone)/gelatin composite electrospun scaffolds with porous crater-like structures for tissue engineering. J Biomed Mater Res A. 2016 Apr;104(4):1017-29. doi: 10.1002/jbm.a.35614]

-          Indicate the geometry and size of the samples that were used for the cell studies.

Response: The following text was included in section 2.5.3, “The geometry and size of the scaffolds used are depicted in Figure 1.”

-          Provide a more detailed and clear explanation regarding how the cell seeding was conducted. From what is currently reported, it seems that the scaffolds were immersed in cell culture media containing cells. If that is the case, why was this approach selected, instead of direct seeding onto the scaffolds? Response: Cells were directly seeded onto the scaffolds. This section was modified to, “A total of 1.2 million cells were seeded directly on top of a scaffold, then completely submerged (approximately 15 mL),

  1. Figure 12:

-    70% ethanol and UV irradiation are disinfection methods, but they do not ensure sterilization (killing 100% of microorganisms). Therefore, it would be extraordinarily difficult to maintain a “clean” culture of the scaffolds for 30 days, as reported by the authors. In this context, why was not cell-metabolic activity assessed? Response: We routinely use this technique to sterilize scaffolds and demonstrated efficacy in a previous publication where cell proliferation and viability was also assessed using MEW scaffolds with long incubation times and using an MTT assay for metabolic assessment. The Hewitt reference have been added to the Discussion section. Ref[13]: Hewitt E, Mros S, McConnell M, Cabral JD, Ali A. Melt-electrowriting with novel milk protein/PCL biomaterials for skin regeneration. Biomed Mater. 2019 Aug 29;14(5):055013. doi: 10.1088/1748-605X/ab3344. PMID: 31318339.

-          Why was not higher magnification used for imaging? Given the above comment, and the fact that Hoechst can bind to damaged DNA as well as DNA of non-mammalian nature, taking images at higher magnification would have allowed to rule out contamination, and also it would have provided a clearer visualization of the integrity of cells attached to the scaffolds.   Response: We have demonstrated cell viability and growth through our MEW PCL scaffolds at higher magnification in previous publications. These references have been added to the Discussion section. Ref [11]Turner PR, Yoshida M, Ali MA, Cabral JD. Melt Electrowritten Sandwich Scaffold Technique Using Sulforhodamine B to Monitor Stem Cell Behavior. Tissue Eng Part C Methods. 2020 Oct;26(10):519-527. doi: 10.1089/ten.TEC.2020.0240. PMID: 32977739. [2] Yoshida M, Turner PR, Ali MA, Cabral JD. Three-Dimensional Melt-Electrowritten Polycaprolactone/Chitosan Scaffolds Enhance Mesenchymal Stem Cell Behavior. ACS Appl Bio Mater. 2021 Feb 15;4(2):1319-1329. doi: 10.1021/acsabm.0c01213. Epub 2021 Jan 13. PMID: 35014483. [13] Hewitt E, Mros S, McConnell M, Cabral JD, Ali A. Melt-electrowriting with novel milk protein/PCL biomaterials for skin regeneration. Biomed Mater. 2019 Aug 29;14(5):055013. doi: 10.1088/1748-605X/ab3344. PMID: 31318339.

  1. In section 3.3.2, the authors state that “This indicated that there was little to no cytotoxic effects introduced by the use of the cPLA molds and that any contaminants that were introduced could be mitigated with standard sterilization procedures.” This is highly inaccurate and misleading, given that cytotoxicity was not measured, and the scaffolds were not sterilized, only disinfected. Please, revise this statement in the updated version of the manuscript. Response: The following sentence was added, “Similar to what we have observed in previously published work by our group growing NHDFs on MEW PCL scaffolds for long term culture, cells were successfully able to proliferate [13] Hewitt E, Mros S, McConnell M, Cabral JD, Ali A. Melt-electrowriting with novel milk protein/PCL biomaterials for skin regeneration. Biomed Mater. 2019 Aug 29;14(5):055013. doi: 10.1088/1748-605X/ab3344. PMID: 31318339.]
  2. Lines 518-519: Neither biocompatibility nor cell proliferation was measured/evaluated. Thus, these words should be removed from the text. Please, revise accordingly.  Response: This has been established in a previous publication as aforementioned above. The focus of this paper was the novelty of shapes created with less focus on biocompatibility as we have established this already in PCL MEW scaffolds. The sentence has been modified to, “Although PCL is not a strongly adherent substrate for cells, the presence of NHDFs on the scaffold after one month indicates that there were no residues introduced in the manufacturing process that impeded growth [[13] Hewitt et al].”
  3. Comments on the Quality of English Language.The writing of the manuscript should be thoroughly revised, as the current version of the document is riddled with syntax and grammar mistakes that impede reading comprehension. Response: The manuscript was thoroughly revised with syntax and grammar corrected throughout.

Reviewer 3 Report

Comments and Suggestions for Authors

The abstract is incompletely written and needs minor revision. The steps of the article should be mentioned first, and then the results should be presented quantitatively and qualitatively. Finally, the most important achievements should be mentioned.

The manuscript needs general writing and grammar editing.

The novelty and purpose of the research should be clearly stated in the abstract and introduction.

The selected keywords such as polycaprolactone are not appropriate and are not mentioned in the abstract.

The introduction is brief. Also, the first paragraphs presented are primarily general and general information. At the end of the introduction, a suitable summary of the importance of the present issue should be provided. Also, discontinuity between paragraphs is evident in most of the introduction. It is suggested to rewrite the introduction.

Use the following resources to deepen the introduction and discussion. Shape memory performance assessment of FDM 3D printed PLA-TPU composites by Box-Behnken response surface methodology. 4D printing of PLA-TPU blends: effect of PLA concentration, loading mode, and programming temperature on the shape memory effect. 4D printing of PET-G via FDM including tailormade excess third shape.

It is suggested to summarize the printing parameters in a table. Also, the process of selecting parameters should be presented. Explain how to choose optimum printing parameters.

How is the print quality checked? Explain more about the printing process. How has the reproducibility of these results been checked?

Some parts of the results are just reporting the results, which require corrections and deepening the analysis and discussion.

The conclusions intend to give significant conclusions based on the results. Critical analysis is required and give the conclusions from the results presented. 

Comments on the Quality of English Language

***

Author Response

  1. The abstract is incompletely written and needs minor revision. The steps of the article should be mentioned first, and then the results should be presented quantitatively and qualitatively. Finally, the most important achievements should be mentioned. Response: The abstract has been rewritten as recommended. The following is highlighted in the manuscript,” We demonstrate for the first time the combination of two additive manufacturing technologies used in tandem, Fused Deposition Modelling (FDM) and Melt-electrowriting (MEW), to increase the range of MEW structures with a focus on creating branched, hollow scaffolds for vascularization. First, computer-aided design (CAD) was used to create branched mold design halves which were then used to FDM print conductive polylactic acid (cPLA) molds. Next, MEW was performed over the top of these FDM cPLA molds using polycaprolactone (PCL), an FDA approved biomaterial, followed by removal of the newly constructed MEW scaffolds from the FDM molds. Lastly, duplicate, complimentary MEW scaffolds were heat melded together by placing two halves together after heating temporarily on a temperature-controlled platform, then pressing them together and allowing them to cool to create hollow scaffold constructs. This hybrid technique permitted direct construction of hollow MEW structures that would otherwise not be possible using MEW alone. The scaffolds then underwent in vitro physical and biological testing. Dynamic mechanical analysis showed the scaffolds had an anisotropic stiffness of 1 MPa or 5 MPa, depending on the direction of the applied stress. And after a month, Normal Human Dermal Fibroblasts (NhDFs) were seen growing on the scaffolds, demonstrating no deleterious effects of MEW scaffolds constructed using FDM cPLA molds. The significant potential of our hybrid additive manufacturing approach to fabricate more complex MEW scaffolds could be applied to a variety of tissue engineering applications, particularly to the field of vascularization.”
  2. The manuscript needs general writing and grammar editing. Response: The manuscript was thoroughly revised with syntax and grammar corrected throughout.
  3. The novelty and purpose of the research should be clearly stated in the abstract and introduction. Response: This has been changed as recommended. The following has been added to the Introduction, “ Therefore, the aim of this research was to use cPLA to create conductive mold halves through FDM, MEW duplicate halves, melt meld them together to create hollow constructs, and observe mold effects on the fabrication of MEW prototype scaffolds as well as characterize their mechanical and biological properties for potential tissue engineering applications.
  4. The selected keywords such as polycaprolactone are not appropriate and are not mentioned in the abstract. Response: Polycaprolactone is now mentioned in the abstract.
  5. The introduction is brief. Also, the first paragraphs presented are primarily general and general information. At the end of the introduction, a suitable summary of the importance of the present issue should be provided. Also, discontinuity between paragraphs is evident in most of the introduction. It is suggested to rewrite the introduction. Response: The Introduction has been rewritten. The following was added in addition to other changes highlighted in the text, “Therefore, the aim of this research was to use cPLA to create conductive, branched mold vessel halves using FDM. MEW halves would then be fabricated on top of the FDM molds, the MEW scaffolds carefully removed from the molds post-print, then melt meld together to create hollow, branched constructs. The cPLA mold effects on the fabrication of MEW prototype scaffolds were determined along with characterizations of their mechanical and biological properties for potential tissue engineering applications.
  6. Use the following resources to deepen the introduction and discussion. To . 4D printing of PLA-TPU blends: effect of PLA concentration, loading mode, and programming temperature on the shape memory effect. 4D printing of PET-G via FDM including tailormade excess third shape. Response: These papers have been included in the Discussion section.
  7. It is suggested to summarize the printing parameters in a table. Also, the process of selecting parameters should be presented. Explain how to choose optimum printing parameters. Response: Tables have been added and the following added to section 2.3,” Optimum print parameters were selected based on surface roughness, mechanical strength, and print fidelity
  8. How is the print quality checked? Explain more about the printing process. How has the reproducibility of these results been checked? Response: The following was added to section 3.2, “ Print fidelity was checked via measurement using digital calipers.”
  9. Some parts of the results are just reporting the results, which require corrections and deepening the analysis and discussion. Response: Results, analysis and discussion have been revised as highlighted in the text.
  10. The conclusions intend to give significant conclusions based on the results. Critical analysis is required and give the conclusions from the results presented.Response: Conclusion has been revised and is highlighted in the text.

Reviewer 4 Report

Comments and Suggestions for Authors

This study titled "Integrating Fused Deposition Modeling and Melt-Electrowriting for Engineering Branched Vasculature" explores the fusion of two 3D printing technologies, Fused Deposition Modeling (FDM) and Melt-Electrowriting (MEW), to engineer complex vasculature structures. While the work presents promising insights, this review report identifies specific areas where corrections and enhancements are needed to further refine the research's potential impact on tissue engineering and regenerative medicine.

Abstract:

1. The abstract provides a clear overview of the study, which investigates the integration of fused deposition modeling (FDM) and melt-electrowriting (MEW) for engineering branched vasculature. The use of conductive polylactic acid FDM-made molds to support MEW scaffolds is introduced. However, there is a minor issue with redundancy in the terminology in the phrase " present a methodology for using conductive polylactic acid fused deposition modeling-made molds," which can be simplified for clarity. Also, it would be helpful to include specific findings or outcomes in the abstract to provide a more concise summary of the results.

2. The use of acronyms such as "NHDF" should be spelled out on their first mention in the abstract to aid reader comprehension.

Introduction:

The introduction provides a comprehensive background on additive manufacturing (AM) technologies, specifically focusing on melt-electrowriting (MEW) and fused deposition modeling (FDM) within the context of tissue engineering. The section effectively explains the principles behind MEW, emphasizing its ability to create highly detailed structures, and highlights the challenges associated with unsupported material in MEW. However, the introduction could be streamlined by avoiding repetition of certain points. For example, the explanation of the challenges related to unsupported material is repeated. Also, the significance of using molds to support scaffold construction is well-established, but the introduction could more directly introduce the research problem and the contribution of this study. Overall, the background information is thorough, and the research objectives are clearly defined.

Materials and Methods:

1. The section briefly mentions the choice of materials such as PLA, cPLA, and PCL. However, it would be beneficial to provide a more comprehensive rationale for selecting these materials over others. What are the specific advantages and limitations of each material in the context of the study?

2. The description of the MEW process is somewhat brief. Expanding on the operational details, such as voltage, temperature, and pressure, would provide a clearer understanding of the experimental setup.

3. Figures and References: Some figures are referred to as "Error! Reference source not found." It is essential to ensure that all figures and references are correctly cited and accessible to readers.

Discussion:

Please discuss potential alternatives to graphene-enhanced conductive PLA (cPLA) that could mitigate brittleness while retaining conductivity, and their feasibility in the Fused Deposition Modeling (FDM) printer.

Conclusions:

How the combination of additive manufacturing technologies can be used for the production of larger, more complex structures. Please provide a roadmap for future research in this area.

Figure and Tables order: Please correct the order/numbering of the Tables. Table 3 is written as Table 1. Also, the order of Figures after Figure 8 is incorrect. Please fix all these.

Figure 12: The number on the scale bar is missing.

Comments on the Quality of English Language

Moderate editing of English language required. Also, please check citing the Figures and references within the text.

Author Response

Abstract:

  1. The abstract provides a clear overview of the study, which investigates the integration of fused deposition modeling (FDM) and melt-electrowriting (MEW) for engineering branched vasculature. The use of conductive polylactic acid FDM-made molds to support MEW scaffolds is introduced. However, there is a minor issue with redundancy in the terminology in the phrase " present a methodology for using conductive polylactic acid fused deposition modeling-made molds," which can be simplified for clarity. Also, it would be helpful to include specific findings or outcomes in the abstract to provide a more concise summary of the results. Response: The abstract has been rewritten as recommended.
  2. The use of acronyms such as "NHDF" should be spelled out on their first mention in the abstract to aid reader comprehension. Response: this was done.
  3. Introduction:

The introduction provides a comprehensive background on additive manufacturing (AM) technologies, specifically focusing on melt-electrowriting (MEW) and fused deposition modeling (FDM) within the context of tissue engineering. The section effectively explains the principles behind MEW, emphasizing its ability to create highly detailed structures, and highlights the challenges associated with unsupported material in MEW. However, the introduction could be streamlined by avoiding repetition of certain points. For example, the explanation of the challenges related to unsupported material is repeated. Also, the significance of using molds to support scaffold construction is well-established, but the introduction could more directly introduce the research problem and the contribution of this study. Overall, the background information is thorough, and the research objectives are clearly defined. Response: the Introduction has been revised.

Materials and Methods:

  1. The section briefly mentions the choice of materials such as PLA, cPLA, and PCL. However, it would be beneficial to provide a more comprehensive rationale for selecting these materials over others. What are the specific advantages and limitations of each material in the context of the study? Response: Highlighted text reflects more detail with regard to material selection.
  2. The description of the MEW process is somewhat brief. Expanding on the operational details, such as voltage, temperature, and pressure, would provide a clearer understanding of the experimental setup. Response: A table with parameters has been included.
  3. Figures and References: Some figures are referred to as "Error! Reference source not found." It is essential to ensure that all figures and references are correctly cited and accessible to readers. Response: This has been corrected.

Discussion:

Please discuss potential alternatives to graphene-enhanced conductive PLA (cPLA) that could mitigate brittleness while retaining conductivity, and their feasibility in the Fused Deposition Modeling (FDM) printer. Response: Other polymers have been suggested and referenced in the Discussion section as highlighted in text.

Conclusions:How the combination of additive manufacturing technologies can be used for the production of larger, more complex structures. Please provide a roadmap for future research in this area. Response: The following was added to the Conclusion, “Future work should involve the exploration of the use of other conducting polymers as FDM molds for MEW scaffold production to produce human-scale tissue constructs featuring the appropriate structural integrity. The combined integration and/or functionalization of bioactives to PCL is key in order to ensure high cell viability and differentiation into appropriate tissue types.”

Figure and Tables order: Please correct the order/numbering of the Tables. Table 3 is written as Table 1. Also, the order of Figures after Figure 8 is incorrect. Please fix all these. Response: Fixed.

Figure 12: The number on the scale bar is missing. Response: Added.

 Comments on the Quality of English Language

Moderate editing of English language required. Also, please check citing the Figures and references within the text. Response: Revised as recommended.

Round 2

Reviewer 2 Report

Comments and Suggestions for Authors

The authors have reasonably addressed the concerns that were initially raised. 

Reviewer 4 Report

Comments and Suggestions for Authors

No further comments are required